# Application of Laser Remelting Technology in the Case of Cultivator Tines

István Domokos * and Sándor Pálinkás *

Department of Mechanical Engineering, Faculty of Engineering, University of Debrecen, 4028 Debrecen, Hungary
* Correspondence: istvan.domokos@eng.unideb.hu (I.D.); palinkassandor@eng.unideb.hu (S.P.)

**Abstract:** The effectiveness of farming relies heavily on the condition of machinery and equipment, as well as maintaining the ideal soil conditions for the desired yields. Soil cultivation tools endure substantial stress and wear, emphasizing the need to study their durability surrounding soil contact. Our research focuses on enhancing the lifespan of worn-out ploughshares through various heat treatment methods and hot metal spraying. By remelting the surface of ploughshares using a flame or laser, we aim to identify the most effective treatment for agricultural production. The improved surface treatment of the furrows in field tillers can significantly cut costs and enhance tillage efficiency. Our preliminary findings suggest that the metal spraying and remelting of nickel alloy hold promise for achieving these goals.

**Keywords:** hot metal powder spraying; agricultural machinery; laser remelting; surface treatment

## 1. Introduction

Coating technologies represent a cornerstone in the advancement of various industries, offering a multitude of applications aimed at enhancing component performance and longevity. From gas turbines to biomedical implants, these coatings play a pivotal role in mitigating wear, corrosion, and thermal stresses, thus contributing to the efficiency and reliability of critical systems [1,2].

In the realm of gas turbines, Thermal Barrier Coatings (TBCs) such as YSZ coatings have emerged as indispensable solutions for preventing superalloy blade failure. Techniques like Atmospheric Plasma Spray (APS) and Cold Gas Spraying (CGS) have further revolutionized the field, offering not only enhanced mechanical and thermal properties but also significant advancements in component lifespan and operational efficiency [2,3].

Based on the research of a German research group [4], we can say that the nuclear industry relies heavily on Thermal Spray Coatings (TSCs) to bolster the corrosion resistance of crucial structures such as drip shields and waste packages. The advent of cold spray technology presents a promising avenue for the deposition of protective coatings, addressing challenges associated with fuel claddings and the mitigation of stress corrosion cracking.

In the aerospace sector of the USA, coatings serve as a frontline defense against the wear, erosion, and oxidation in engine components. Techniques like Atmospheric Plasma Spray (APS) enable the precise and uniform application of coatings, thereby ensuring optimal performance even in the most demanding operating environments [5].

In India, biomedical applications leverage advanced coating technologies to promote the biocompatibility and osseointegration of implants. Highly crystalline nano hydroxyapatite (HA) coatings, deposited using methods like inductively coupled radio frequency (RF) plasma spray, showcase promising results in enhancing the biological response of implant materials [6,7].

Moreover, the automotive industry has explored the potential of coating technologies to improve the wear resistance and durability of components. Amorphous iron-based

coatings, applied using powder and wire flame spray techniques, demonstrate superior performance characteristics, thereby enhancing the reliability of automotive systems.

Thermal spraying is a process that allows for the deposition of molten, semi-molten, or solid particles onto a substrate, enhancing the performance and functionality of components. As a result, it finds applications across various industries, including the automotive, energy, and medical sectors. The process is highly versatile, and almost any material that melts and does not decompose can be utilized. An important advantage is that it does not require significant heat input, allowing even materials with very high melting points to be applied to components without altering their properties. Additionally, it offers us the ability to replace worn or deteriorated coatings without modifying the properties or dimensions of the parts. Through a wide range of techniques and coating materials, thermal spraying can enhance desirable characteristics without thermally affecting the components [1].

Agricultural activities are fundamentally determined by the condition of power and soil cultivation machinery. The active components of machinery used in soil cultivation are subjected to significant wear during operation (Figure 1). Several research groups in the international literature have focused on improving the wear resistance of soil cultivation elements [8–13]. Achieving the desired yield is greatly influenced by the establishment and maintenance of proper soil conditions. Cultivator blades, therefore, wear out very quickly, forcing farmers to continuously repair and replace them [14–16]. The aim of this research is to increase the wear resistance of these blades, thereby significantly extending their lifespan. Although this solution may be more expensive initially, it quickly pays off due to reduced maintenance and operating costs. In this study, we aim to answer the question of what wear results can be obtained when blade tips are produced using hot metal spraying, and how effectively the layer applied by hot metal spraying can protect their surface from mechanical impacts. During hot metal powder spraying, the powder is sprayed in a semi-molten state onto the preheated workpiece for fusion purposes [17–22]. Alloys are bonded to the base metal by diffusion. Figure 2 shows the formation of a molten layer of hot metal spray on the surface of stainless steel, under 500× magnification. The sprayed layer's diffusion zone and the base metal can be seen. In our research, half of the blades treated with hot metal spraying were remelted with a flame, and the other half with a laser.

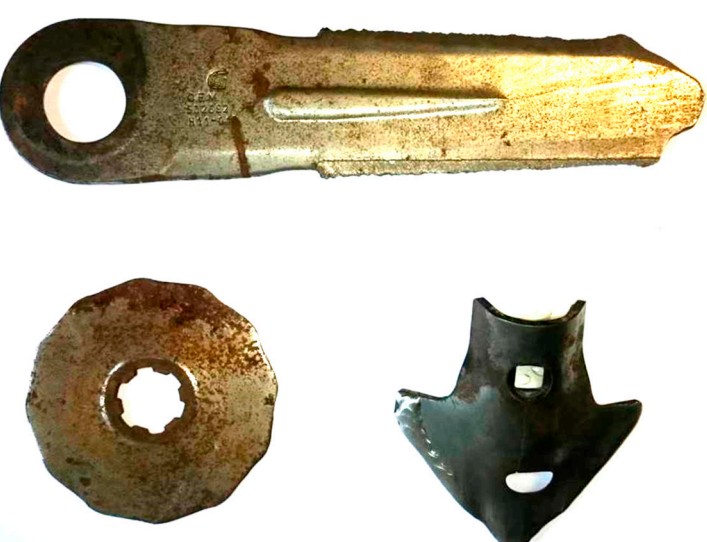

**Figure 1.** Worn agricultural machine parts. Corn-stalk crushing knifes; disc of corncob breaking adapter; cultivator tine [16].

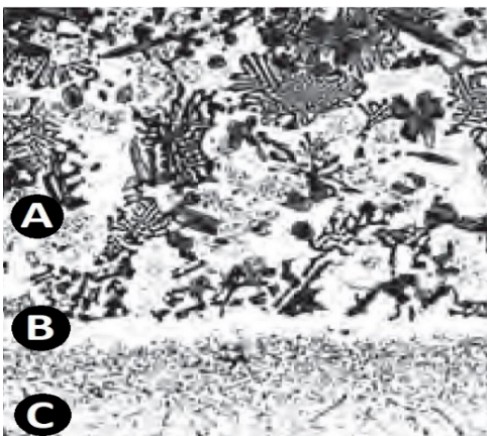

**Figure 2.** The formation of a molten layer of hot metal spray on the surface of stainless steel, 500× magnification: (**A**) sprayed layer, (**B**) diffusion zone, (**C**) base metal [23].

## 2. Materials and Methods

### 2.1. Preparation of Samples for Soil Cultivation

The hot metal powder spraying of the experimental samples was conducted at the welding laboratory of the Department of Mechanical Engineering, Faculty of Engineering, University of Debrecen. We prepared 8 hot-metal-sprayed cultivator tines for the experiment. The chosen raw tine is shown in Figure 3. We used a C60-grade hot-rolled steel sheet for the cultivator tines. We have summarized the chemical composition of C60 steel in Table 1.

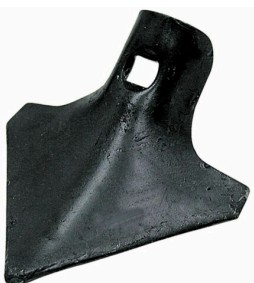

**Figure 3.** The raw cultivator tine.

**Table 1.** Chemical composition % of C60 steel [24].

| Cr + Mo + Ni = max 0.63 | | | | | | | |
|---|---|---|---|---|---|---|---|
| C | Si | Mn | Ni | P | S | Cr | Mo |
| 0.57–0.65 | max 0.4 | 0.6–0.9 | max 0.4 | max 0.045 | max 0.045 | max 0.4 | max 0.1 |

The first 4 tines (samples 1–4) were re-melted using a flame. The remelting process was carried out using an acetylene–oxygen gas mixture. The second set of 4 tines (samples 5–8) were also first treated by the aforementioned hot metal spray fusing, and then they were remelted using a laser. In the case of two additional pieces (sample 9 and 10), laser deposition welding was used. The cladding, specifically laser cladding, was performed using Höganäs 60% Tungsten Carbide powder. This process was conducted by the laser technology facility of BuBen Laser Budai Benefit Ltd (Halásztelek, Hungary). The cultivator tines prepared for the experiment are summarized in Table 2.

During the process, the workpiece was heated to a temperature of approximately 250–300 °C. This is called the wetting process. The correct choice of raw material is important, so that, during metal powder spraying, a good enough quality bond is created between the applied metal powder and the workpiece. During metal powder spraying (Figure 4), the temperature of the workpieces is very high, therefore, the nature of their

cooling is important to ensure hardness. In our case, the workpieces were cooled in free air. For the experiment, 10 cultivator tines were available.

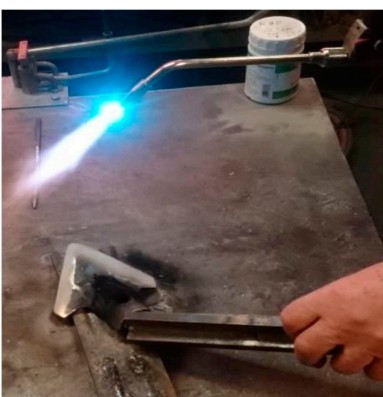
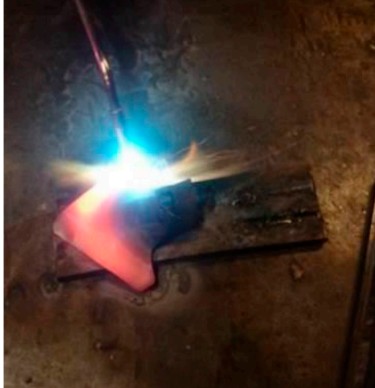

**Figure 4.** Application of a layer via hot metal powder spraying (source: self-made).

**Table 2.** Surface treatment processes for cultivator tines.

| Number of Samples | Applied Coating Deposition Technology | Method of Remelting |
|---|---|---|
| 1 | hot metal spraying with Eutalloy® 10009 (BoroTec) (Castolin Eutectic, Lausanne, Switzerland) alloy powder | Flame |
| 2 | hot metal spraying with N 60 Mogul alloy powder | Flame |
| 3 | hot metal spraying with N 60 Mogul alloy powder | Flame |
| 4 | hot metal spraying with N 60 Mogul alloy powder | Flame |
| 5 | hot metal spraying with N 60 Mogul alloy powder | Laser |
| 6 | hot metal spraying with N 60 Mogul alloy powder | Laser |
| 7 | hot metal spraying with N 60 Mogul alloy powder | Laser |
| 8 | hot metal spraying with 50% N 60 Mogul + 50% Tungsten Carbide | Laser |
| 9 | laser cladding with Höganäs 60% Tungsten Carbide | |
| 10 | laser cladding with Höganäs 60% Tungsten Carbide | |

The main advantage of laser remelting is that, in contrast to remelting with a flame, we can control the temperature and the speed of the laser very precisely, as a result of which, the heat treatment takes place with a high degree of control. So, reduced brittleness in the raw material and a reduced dilution of the sprayed layer are expected, meaning the transition zone between the base material and the sprayed layer is ideally mixed, thus creating a cohesive bond. The finished experimental samples are shown in Figures 5 and 6.

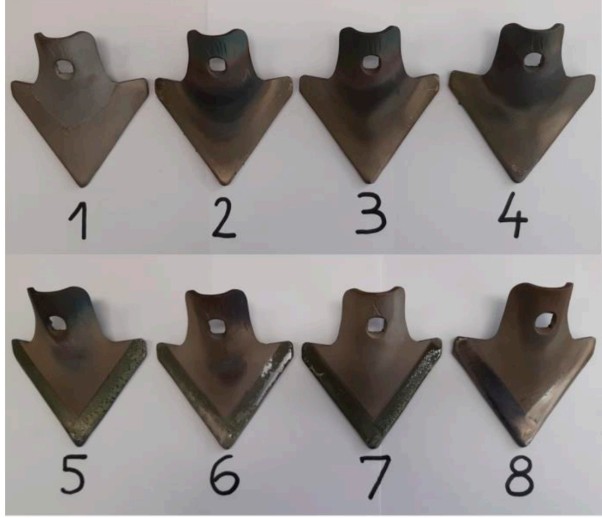

**Figure 5.** The hot-metal-sprayed tines (source: self-made).

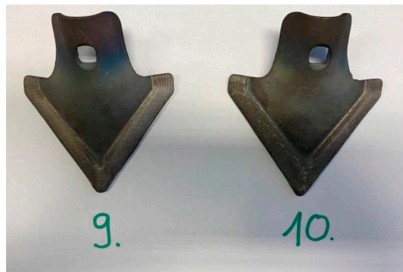

**Figure 6.** The samples made through laser cladding (source: self-made).

*2.2. Preparation of Samples for Material Analyses and Ultrasonic Hardness Testing*

For the metallographic and ultrasonic hardness examinations, we created 4 test samples (A, B, C, D), the surfaces of which were treated with hot metal spraying and remelted using a flame on the left side (side "1") and laser on the right side (side "2") (Figure 7).

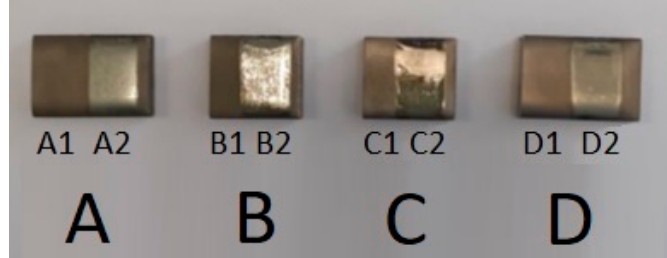

**Figure 7.** Samples remelted with a flame on the left and with a laser on the right (source: self-made).

The material of the cultivator tines is a C60-grade hot-rolled steel sheet. At the beginning of the experiment, the main aim of the investigation was to examine the effect of laser re-melting on the sprayed layer. For this purpose, we utilized the SAU-TER HO ultrasonic hardness tester (Figure 8). The SAUTER HO measures the hardness of a test specimen using a Vickers diamond tip, which is pressed onto the surface of the test specimen with a specified force. Subsequently, the tip is vibrated by ultrasound. The applied frequency was 7 MHz [25]. The minimum measurable thickness the ultrasonic hardness tester can use for tests is 0.75 mm. For most samples, the thickness was 1.8 mm, while for sample C2 it was 0.8 mm, indicating that the inspected volume contained the treated zones. The device was set for measuring minimal thickness. Due to its speed, this method is advantageous compared to conventional surface hardness testing. The device calculates hardness values from the attenuation of ultrasound.

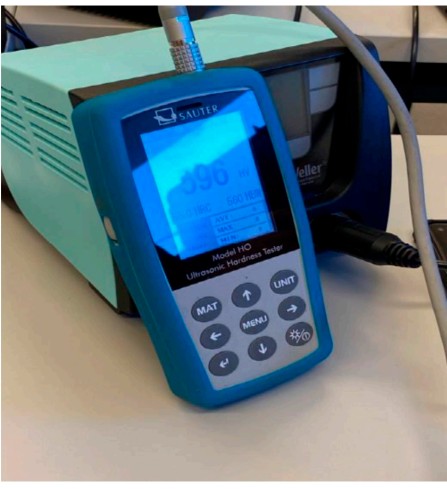

**Figure 8.** SAUTER HO ultrasonic hardness tester (source: self-made).

During the hot metal spraying process, different layers were applied to steel plates of the same material quality as the cultivator tine, thereby creating test pieces for the purpose of examining the transitional layer formed between the applied layer and the base material. The remelting of the test samples was carried out using a flame and laser. The composition of the layers and the method of re-melting are detailed in Table 3. The finished samples prepared for microscopic analyses are shown in Figure 9.

**Table 3.** Test samples.

| Number of the Tine | Applied Coating Deposition Technology | Method of Remelting |
|---|---|---|
| A1 | Hot metal spraying with Deloro 60 alloy powder | Flame |
| B1 | Hot metal spraying with N 40 Mogul alloy powder | Flame |
| C1 | Hot metal spraying with N 50 Mogul alloy powder | Flame |
| D1 | Hot metal spraying with N 60 Mogul alloy powder | Flame |
| A2 | Hot metal spraying with Deloro 60 alloy powder | Laser |
| B2 | Hot metal spraying with N 40 Mogul alloy powder | Laser |
| C2 | Hot metal spraying with N 50 Mogul alloy powder | Laser |
| D2 | Hot metal spraying with N 60 Mogul alloy powder | Laser |

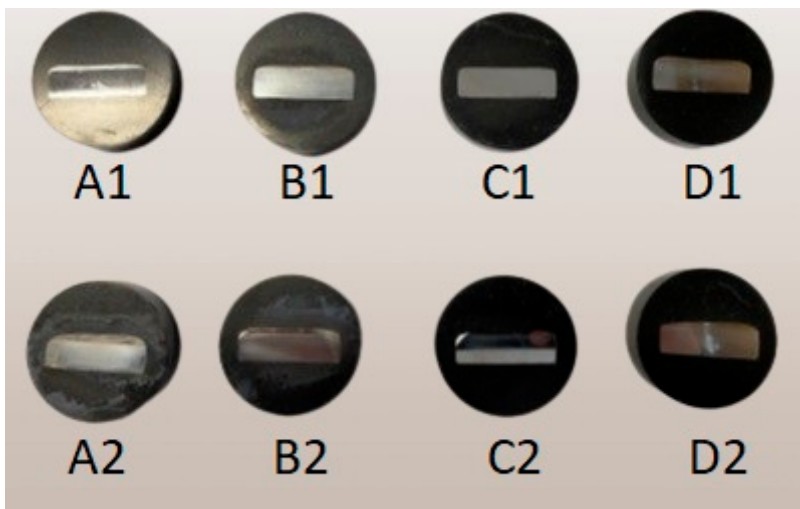

**Figure 9.** The samples prepared for microscopic analysis (source: self-made).

For a better understanding, we have summarized the composition of the metal powders used in Tables 4–7.

**Table 4.** The composition of the used Deloro 60 metal powder.

| Nominal Composition [mass%] of | | | | | |
|---|---|---|---|---|---|
| Ni | Cr | Si | B | C | Fe |
| Base | 7.5 | 4 | 1.5 | 0.25 | 3 |

**Table 5.** The composition of the used Mogul N 40 powder.

| Nominal Composition of | | | | | |
|---|---|---|---|---|---|
| Ni | Cr | Si | B | C | Fe |
| Base | 7.5 | 3.6 | 1.65 | 0.25 | 1.3 |

**Table 6.** The composition of the used Mogul N 50 metal powder.

| Nominal Composition | | | | | |
|---|---|---|---|---|---|
| Ni | Cr | Si | B | C | Fe |
| Base | 14 | 3.7 | 2.5 | 0.5 | 4 |

**Table 7.** The composition of the used Mogul N 60 metal powder.

| Nominal Composition | | | | | |
|---|---|---|---|---|---|
| Ni | Cr | Si | B | C | Fe |
| Base | 15 | 4.3 | 3.1 | 0.75 | 4.5 |

## 3. Results and Evaluations

### 3.1. Result of Ultrasonic Hardness Testing

The measurement was repeated three times for all samples. The results are shown in Table 8. Based on the obtained results, it can be concluded that laser re-melting significantly increased the hardness of the sprayed surface in the cases of samples A, B, and C.

**Table 8.** The results of the ultrasonic hardness test.

| Number of Samples | Remelted | Hardness (HRC) | | | Average (HRC) |
|---|---|---|---|---|---|
| A1 | flame | 49.1 | 49.4 | 54.1 | 50.9 |
| A2 | laser | 56.9 | 55.8 | 58.9 | 57.2 |
| B1 | flame | 38.7 | 39.2 | 37.5 | 38.5 |
| B2 | laser | 48.6 | 48.7 | 48.6 | 48.6 |
| C1 | flame | 48.7 | 43.8 | 45.5 | 46 |
| C2 | laser | 62.5 | 63.8 | 64 | 63.4 |
| D1 | flame | 57.6 | 58.9 | 59.5 | 58.7 |
| D2 | laser | 58.6 | 58.8 | 58.9 | 58.8 |

### 3.2. Microscopic Analyses

Finally, the prepared test samples and their microstructures were examined using a Zeiss Smartzoom 5 optical microscope (Oberkochen, Germany). We were interested in understanding how deeply the laser remelting can penetrate the sprayed layer, as well as what microstructural changes occur during the laser re-melting process. We aimed to achieve the most suitable magnification to ensure the visibility of the base material, the sprayed layer, and the diffusion zone.

During our microscopic examinations, we compared the microstructures of the tines treated with flame re-melting and laser re-melting. Upon evaluation, it was observed that, in each sample, the base material, the transitional diffusion zone, and the sprayed layer are clearly distinguishable. Additionally, in the case of samples remelted with the laser, the remelted layer is clearly discernible.

One of the advantages of the Zeiss microscope is that it is possible to analyze the surface of samples and take very detailed surface images. In Figure 10, it can be observed that the layer remelted by flame on the left side and by laser on the right side is clearly separated.

In Figures 11–14, the base material, the transitional diffusion zone, and the sprayed layer are distinctly visible. Furthermore, the images also show that there is no mixing (dilution) with the base material.

During the evaluation of these microscopic images, we found that the penetration depth of the laser remelting can be obviously observed on samples A2, B2, and D2. In sample C2, the laser completely penetrated the sprayed layer because the layer was too thin. The thickness of the layer on the surface of sample C2 was 0.8 mm, while for the other samples it was 1.8 mm thick. Since the laser completely melted sample C2 and melted the other samples to a depth of 0.8 mm, we conclude that the laser penetration depth at the applied settings (temperature was 1150 °C, feed was 3 mm/s) was 0.8 mm.

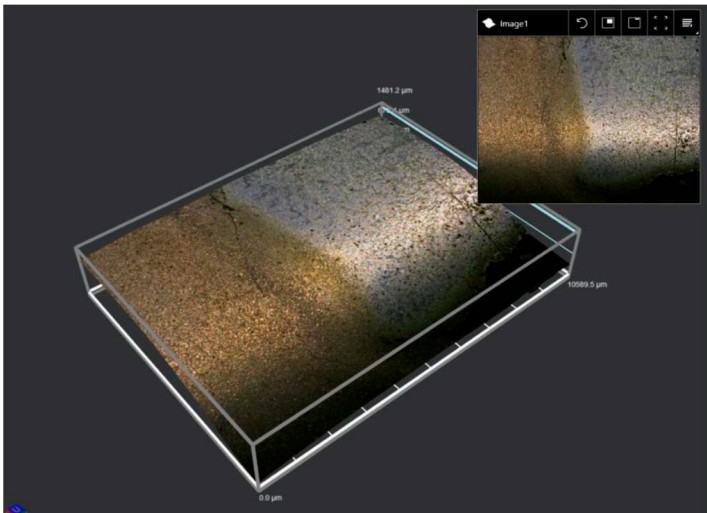

**Figure 10.** Surface image of the sprayed layer; the left side was remelted by flame and the right side was remelted by laser.

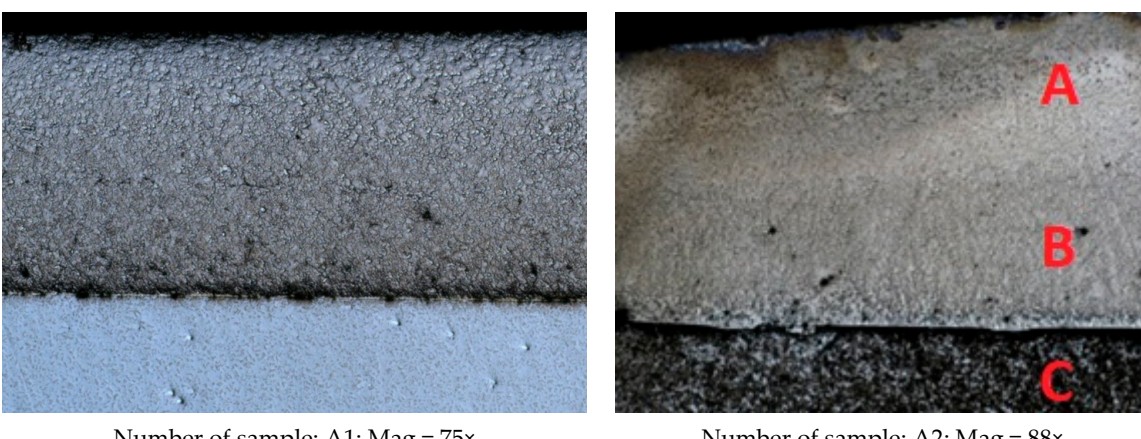

Number of sample: A1; Mag = 75×    Number of sample: A2; Mag = 88×

**Figure 11.** Comparison of microscopic images of samples sprayed with Deloro 60 alloy powder and remelted with a flame or laser. (**A**) Layer remelted by laser, (**B**) diffusion zone, (**C**) base metal.

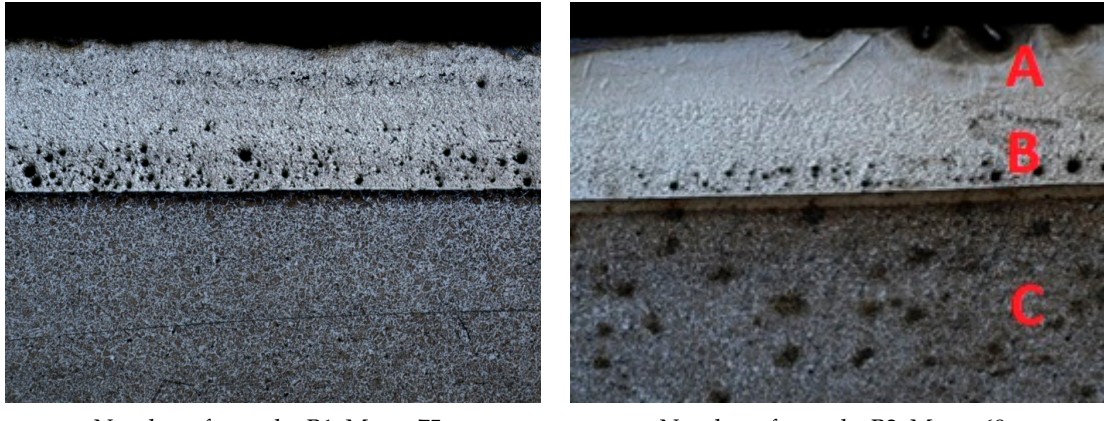

Number of sample: B1; Mag = 75×    Number of sample: B2; Mag = 69×

**Figure 12.** Comparison of microscopic images of samples sprayed with N40 Mogul alloy powder and remelted with a flame or laser. (**A**) Layer remelted by laser, (**B**) diffusion zone, (**C**) base metal.

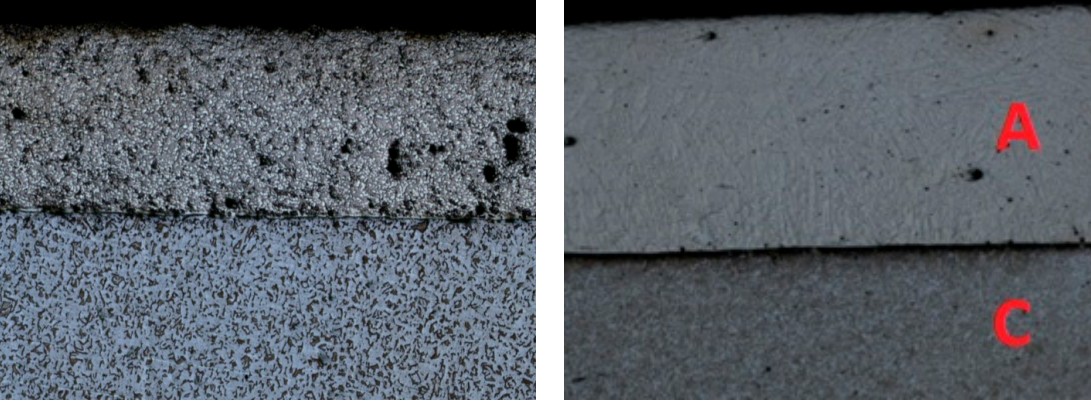

Number of sample: C1; Mag = 116×          Number of sample: C2; Mag = 156×

**Figure 13.** Comparison of microscopic images of samples sprayed with N50 Mogul alloy powder and remelted with a flame or laser. (**A**) Layer remelted by laser, (**C**) base metal.

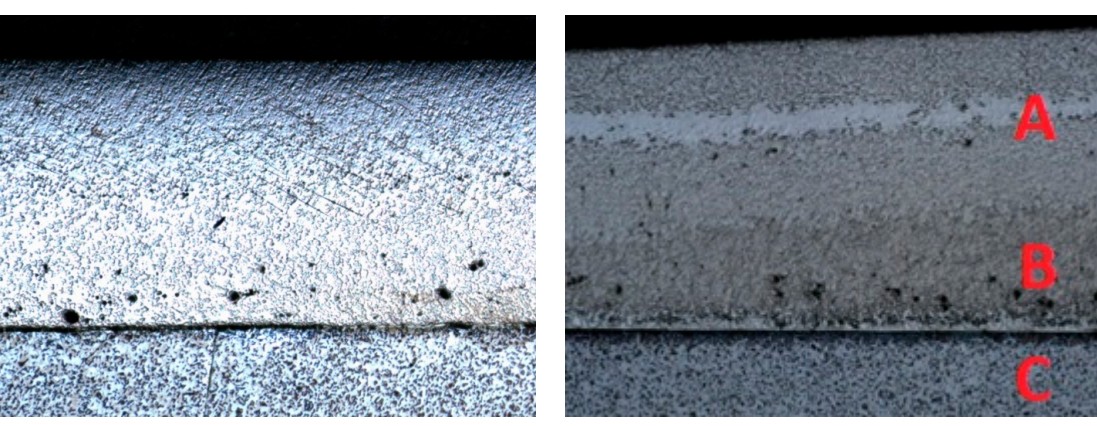

Number of sample: D1; Mag = 52×          Number of sample: D2; Mag = 85×

**Figure 14.** Comparison of microscopic images of samples sprayed with N60 Mogul alloy powder and remelted with a flame or laser. (**A**) Layer remelted by laser, (**B**) diffusion zone, (**C**) base metal.

## 4. Conclusions

The interesting aspect of this paper is that, after the initial hot metal powder spraying, we used laser remelting next to remelting with a flame. During our research, we prepared two experiments. In the first experiment, we prepared eight tines for field testing purposes using different types of powder, and then we remelted them with a flame or laser, and, in addition, two samples were created by laser cladding them with tungsten carbide. Since the testing of these can only be carried out at certain times of the year, the examination of these 10 tines will take place later, and we plan to publish the results after the agricultural season.

The second part of our investigation involved preparing samples for material analysis and ultrasonic hardness testing. We treated the samples with four types of powders: Deloro 60, Mogul N40, Mogul N50, and Mogul N60. From these samples, we created pairs; one side of the sample was remelted with a flame, while the other side was remelted with a laser. In total, eight types of coatings were prepared for ultrasonic hardness testing and microscopic analysis. The surface of the treated samples was then analyzed using an ultrasonic hardness test and an optical microscope. During our microscopic examinations, we compared the microstructure of the samples subjected to flame remelting after hot metal spraying to the samples subjected to laser remelting.

Our research has achieved its original goal. In this paper, we have proven that surfaces treated with hot metal spraying will be harder after laser remelting than flame melting. During this evaluation, we found that the base material, the transitional diffusion zone,

and the scattered layer were clearly visible on all samples. Furthermore, during the laser remelting experiments, it was possible to find the parameters at which there is no dilution between the laser-remelted layer and the base material.

**Author Contributions:** Methodology, I.D. and S.P.; Investigation, I.D.; Writing—original draft, I.D.; Supervision, S.P.; Project administration, I.D. All authors have read and agreed to the published version of the manuscript.

**Funding:** This research received no external funding.

**Institutional Review Board Statement:** Not applicable.

**Informed Consent Statement:** Not applicable.

**Data Availability Statement:** Data are contained within the article.

**Acknowledgments:** We would like to thank our colleagues who contributed to the creation of this scientific work. Special thanks to Márton Lévai, engineering teacher, and András Molnár, guest professor, for their help in carrying out the creation of the samples via hot metal powder spray fusing. We are very grateful to BuBen Laser Budai Benefit Kft. for their help during the laser remelting of the samples.

**Conflicts of Interest:** The authors declare no conflict of interest.

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
