# Peer review of "Application of Laser Remelting Technology in the Case of Cultivator Tines"

_coatings, doi:10.3390/coatings14050637_

Round 1

Reviewer 1 Report

Comments and Suggestions for Authors

Application of laser remelting technology in case of cultivator tines

The manuscript deals with the improvement of the surface properties of tiles for agricultural applications. The authors coat the tiles with hot metal spraying using different powders alloys and remelt the coating using flame or laser. Microscopical observations and hardness evaluation of the samples are described.

The analyses conducted are interesting for the readership of Coatings and appealing to other readers. Nevertheless, the document is not well written and organized, and the scientific descriptions and discussion should be dramatically improved.

·         Deepen the scientific description in the introduction providing also quantitative pieces of information.

·         Please, provide more information regarding the tines treated: raw material, composition, geometry.

·         If I understand well, the samples 1-10 reported in Table 1 have not been analyzed. Indeed, the numbering of the samples reported in Table 2 is not consistent with the numbering of the samples reported in Table 1. This is extremely confusing. What is the reason for the production of the samples 1-10?

·         What is the difference among samples 2, 3, and 4 (Table 1)? What is the difference among samples 5, 6, and 7 (Table 1)?

·         The authors should illustrate the meaning of N40, N50, N60.

·         The inspection and analysis methods should be described in the Materials and Methods section.

·         The setting of the ultrasonic hardness testing system (Frequency, amplitude, wave shape) determines the inspected volume. Did the authors check if the inspected volumes involve only the treated zones? Until which depth? Was this method used previously for this application and why it is preferable to a common surface hardening testing?

·         Figure 9 is not useful for the discussion. It would be much better to report more details about the observed surfaces.

·         The comments on the achieved results and the scientific discussion are extremely poor. 

Author Response

We thank to the Reviewer for the interest in our work and for the helpful comments that will greatly improve the manuscript. We have tried to do our best to respond to the comments raised.

Point 1 -              Deepen the scientific description in the introduction providing also quantitative pieces of information.

Response 1: Thank you for the feedback! I attempted to expand the scientific description. I utilized 9 sources to present the current state of powder spraying and coating. As a result, I enhanced the introduction to the topic, aiding in better understanding.

  1. Fazekas L. (2011). Cold spray deposited surface layer properties and their relationship to technology Szent István University, GödöllÅ‘, Hungary
  2. Shin, D.; Gitzhofer, F.; Moreau, C. Development of Metal Based Thermal Barrier Coatings (MBTBCs) for Low Heat Rejection Diesel Engines. Thermal Spray 2005: Proceedings from the International Thermal Spray Conference 2005, doi:10.31399/asm.cp.itsc2005p0915.
  3. Gaur, U.P.; Kamari, E. Applications of Thermal Spray Coatings: A Review. Journal of Thermal Spray and Engineering 2024, 4, 106–114, doi:10.52687/2582-1474/405.
  4. Vaßen, R.; Kaßner, H.; Stuke, A.; Hauler, F.; Hathiramani, D.; Stöver, D. Advanced Thermal Spray Technologies for Applications in Energy Systems. Surface and Coatings Technology 2008, 202, 4432–4437, doi:10.1016/j.surfcoat.2008.04.022.
  5. Yeom, H.; Sridharan, K. Cold Spray Technology in Nuclear Energy Applications: A Review of Recent Advances. Annals of Nuclear Energy 2021, 150, 107835, doi:10.1016/j.anucene.2020.107835.
  6. Singh, H.; Kumar, R.; Prakash, C.; Singh, S. HA-Based Coating by Plasma Spray Techniques on Titanium Alloy for Orthopedic Applications. Materials Today: Proceedings 2022, 50, 612–628, doi:10.1016/j.matpr.2021.03.165.
  7. Prashar, G.; Vasudev, H. Thermal Sprayed Composite Coatings for Biomedical Implants: A Brief Review. Journal of Thermal Spray and Engineering 2020, 2, 50–55, doi:10.52687/2582-1474/213.
  8. Bitay, E. ). Laser surface treatment and modeling (Lézeres Felületkezelés És Modellezés) Műszaki Tudományos Füzetek 2007, doi:10.36242/mtf-04.
  9. Chikarakara, E.; Aqida, S.; Brabazon, D.; Naher, S.; Picas, J.A.; Punset, M.; Forn, A. Surface Modification of HVOF Thermal Sprayed WC–CoCr Coatings by Laser Treatment. International Journal of Material Forming 2010, 3, 801–804, doi:10.1007/s12289-010-0891-0.

Point 2: Please, provide more information regarding the tines treated: raw material, composition, geometry.

Response 2:

Thank you for the acknowledgment! In section 2.1, I supplemented the article with information about the cultivator tine and the chemical composition of C60 steel.

Point 3: If I understand well, the samples 1-10 reported in Table 1 have not been analyzed. Indeed, the numbering of the samples reported in Table 2 is not consistent with the numbering of the samples reported in Table 1. This is extremely confusing. What is the reason for the production of the samples 1-10?

Response 3:

You're absolutely right. I didn't clearly distinguish that we prepared two types of samples. We prepared the shovels for field testing, but the test specimens were intended for ultrasonic hardness testing and microscopic examination. I've corrected this oversight with care. I apologize for the earlier omission and thank you for bringing it to my attention. I changed the labels to make them clear.

Point 4: What is the difference among samples 2, 3, and 4 (Table 1)? What is the difference among samples 5, 6, and 7 (Table 1)?

Response 1: There is no difference between samples 2, 3, 4, and samples 5, 6, 7; they were produced using the same technology. These are designed for field testing, as mentioned in the article, which will be conducted later this year. To conduct the test credibly, we have prepared 3 pieces each of the two most important test specimens

Point 5:  The authors should illustrate the meaning of N40, N50, N60.

Response 5: I supplemented the text with Table 4, Table 5, Table 6, and Table 7, which summarize the composition of various metal powders.

Point 6: The setting of the ultrasonic hardness testing system (Frequency, amplitude, wave shape) determines the inspected volume. Did the authors check if the inspected volumes involve only the treated zones? Until which depth? Was this method used previously for this application and why it is preferable to a common surface hardening testing?

Response 6: The SAUTER HO measures the hardness of the test specimen using a Vickers diamond tip, which is pressed onto the surface of the test specimen with a specified force. Subsequently, the tip is vibrated by ultrasound. The applied frequency was 7 MHz . The minimum measurable thickness of the ultrasonic hardness tester is 0.75 mm. For most samples, the thickness was 1.8 mm, and for sample C2, it was 0.8 mm, indicating that the inspected volume contained the treated zones. The device was set for measuring minimal thickness. Due to its speed, this method is advantageous compared to conventional surface hardness testing. The device calculates the hardness value from the attenuation of ultrasound.

Point 7: Figure 9 is not useful for the discussion. It would be much better to report more details about the observed surfaces

Response 7: I’m agree. I deleted from the paper. Thank you!

Point 8: The comments on the achieved results and the scientific discussion are extremely poor. 

Response 8: Thank you for the feedback. I improved it!

Reviewer 2 Report

Comments and Suggestions for Authors

1. Very poor prepared critical review of world literature. The authors of the article did not describe what is being done in this field in the world and what they propose in their article.

                      2. The introduction to the article is poorly written. It should be significantly expanded to include a global literature review.   3. In the introduction to the article, the authors cited two photos - but there is no detailed description of them. Please interpret them thoroughly.   4. Generally, there are a lot of photos in the article, but without their detailed description - in the text, or preferably in photos using e.g. text fields.   5. Item No. 3 is very "old" - from 1986, or the authors do not have a more recent study. "Hartmann V., Felker J., Kalmár V., Horváth G.; MezÅ‘gazdasági gépalkatrészek felújitása, MezÅ‘gazdasági Kiadó, Budapest, 1986; 198 pp. 87-92".   6. In the list of bibliography - item no. 7, there is no year of publication or date of access. István, D.; Sándor, P.; Lajos, F.; András, M. Lézeres újraolvasztási technológia alkalmazása kultivátorkapák esetén Application of laser 206 remelting technology in case of cultivator tine; https://mecheng.unideb.hu   7. In the bibliography list, some items are in Hungarian and should be in English.   8. Very poorly prepared summary - conclusions to the article, they should be clarified and written down what the authors of the article plan next - what research.   9. We do not provide academic degrees next to our names.

Author Response

Response to Reviewer 2 Comments

We thank to the Reviewer for the interest in our work and for the helpful comments that will greatly improve the manuscript. We have tried to do our best to respond to the comments raised.

Point 1 -               . Very poor prepared critical review of world literature. The authors of the article did not describe what is being done in this field in the world and what they propose in their article.

Response 1: Thank you for the feedback! I attempted to expand the scientific description. I utilized 9 sources to present the current state of powder spraying and coating. As a result, I enhanced the introduction to the topic, aiding in better understanding.

  1. Fazekas L. (2011). Cold spray deposited surface layer properties and their relationship to technology Szent István University, GödöllÅ‘, Hungary
  2. Shin, D.; Gitzhofer, F.; Moreau, C. Development of Metal Based Thermal Barrier Coatings (MBTBCs) for Low Heat Rejection Diesel Engines. Thermal Spray 2005: Proceedings from the International Thermal Spray Conference 2005, doi:10.31399/asm.cp.itsc2005p0915.
  3. Gaur, U.P.; Kamari, E. Applications of Thermal Spray Coatings: A Review. Journal of Thermal Spray and Engineering 2024, 4, 106–114, doi:10.52687/2582-1474/405.
  4. Vaßen, R.; Kaßner, H.; Stuke, A.; Hauler, F.; Hathiramani, D.; Stöver, D. Advanced Thermal Spray Technologies for Applications in Energy Systems. Surface and Coatings Technology 2008, 202, 4432–4437, doi:10.1016/j.surfcoat.2008.04.022.
  5. Yeom, H.; Sridharan, K. Cold Spray Technology in Nuclear Energy Applications: A Review of Recent Advances. Annals of Nuclear Energy 2021, 150, 107835, doi:10.1016/j.anucene.2020.107835.
  6. Singh, H.; Kumar, R.; Prakash, C.; Singh, S. HA-Based Coating by Plasma Spray Techniques on Titanium Alloy for Orthopedic Applications. Materials Today: Proceedings 2022, 50, 612–628, doi:10.1016/j.matpr.2021.03.165.
  7. Prashar, G.; Vasudev, H. Thermal Sprayed Composite Coatings for Biomedical Implants: A Brief Review. Journal of Thermal Spray and Engineering 2020, 2, 50–55, doi:10.52687/2582-1474/213.
  8. Bitay, E. ). Laser surface treatment and modeling (Lézeres Felületkezelés És Modellezés) Műszaki Tudományos Füzetek 2007, doi:10.36242/mtf-04.
  9. Chikarakara, E.; Aqida, S.; Brabazon, D.; Naher, S.; Picas, J.A.; Punset, M.; Forn, A. Surface Modification of HVOF Thermal Sprayed WC–CoCr Coatings by Laser Treatment. International Journal of Material Forming 2010, 3, 801–804, doi:10.1007/s12289-010-0891-0.

Point 2:  In the introduction to the article, the authors cited two photos - but there is no detailed description of them. Please interpret them thoroughly.

Response 2:

Thank you for the acknowledgment! I supplemented the text with the requested information, ad I made the title of the images more understandable:

 Agricultural activities are fundamentally determined by the condition of power and soil cultivation machinery. The active components of machinery used in soil cultivation are subjected to significant wear during operation (Figure 1). Several research groups in the international literature are focused on improving the wear resistance of soil cultivation elements [11-13]. Achieving the desired yield is greatly influenced by the establishment and maintenance of proper soil conditions. Cultivator blades, therefore, wear out very quickly, forcing farmers to continuously repair and replace them [14-16]. The aim of this research is to increase the wear resistance of these blades, thereby significantly extending their lifespan. Although this solution may be more expensive initially, it quickly pays off due to reduced maintenance and operating costs. In this study, we aim to answer the question of what wear results can be obtained if the blade tips are produced using hot metal spraying, and how effectively the layer applied by hot metal spraying can protect the surface from mechanical impacts. During hot metal powder spraying, the powder is sprayed in a semi-molten state onto the preheated workpiece for fusion purposes [17-22]. Alloys are bonded to base metal by diffusion. The figure 2 shows the formation of a molten layer of hot metal spray on the surface of stainless steel, in case 500x magnification. It can be observed the sprayed layer diffusion zone and the base metal. In our research, half of the blades treated with hot metal spraying were remelted by flame, and the other half by laser.

Point 3 Generally, there are a lot of photos in the article, but without their detailed description - in the text, or preferably in photos using e.g. text fields

Response 3:

You are right! I tried to clarify what we can see at the figures, to make them more understandable.

Point 4:  Item No. 3 is very "old" - from 1986, or the authors do not have a more recent study. "Hartmann V., Felker J., Kalmár V., Horváth G.; MezÅ‘gazdasági gépalkatrészek felújitása, MezÅ‘gazdasági Kiadó, Budapest, 1986; 198 pp. 87-92".  

Response 4: We found a newer research, and we changed it!  

Point 5: In the list of bibliography - item no. 7, there is no year of publication or date of access. István, D.; Sándor, P.; Lajos, F.; András, M. Lézeres újraolvasztási technológia alkalmazása kultivátorkapák esetén Application of laser 206 remelting technology in case of cultivator tine; https://mecheng.unideb.hu 

Response 5: Thank You. It is our work, but it is not necessary here, so I deleted it from the list.

Point 6  In the bibliography list, some items are in Hungarian and should be in English.

Response 6: Thank you! We translated them to English!

Point 7: Very poorly prepared summary - conclusions to the article, they should be clarified and written down what the authors of the article plan next - what research.

Response 7: We agree! We improved the discussion!

“The interesting aspect of this paper that after the initial hot metal powder spraying, we used laser remelting next to the remelting with flame. During our research, we prepared two experiments. In the first experiment, we prepared 8 tines for field testing purposes, with different types of powder, and then we remelted them with a flame or laser and in addition two samples were applied by laser cladding with tungsten carbide. Since the testing of these can only be carried out at certain times of the year, the examination of these 10 tines will take place later, and we plan to publish the results after the agricultural season.”

Point 8: We do not provide academic degrees next to our names.

Response 8: Thank you! I deleted them!

Round 2

Reviewer 1 Report

Comments and Suggestions for Authors

Application of laser remelting technology in case of cultivator tines

The authors effectively revised and improved the manuscript following the peer-review comments. In the referee’s opinion, it is suitable for publication in Coatings.